# Noether's Learning Dynamics:
# Role of Symmetry Breaking in Neural Networks

**Hidenori Tanaka**[1*] , **Daniel Kunin**[2]
[1]Physics & Informatics Laboratories, NTT Research, Inc., Sunnyvale, CA, USA
[2]Stanford University, Stanford, CA, USA

## Abstract

In nature, symmetry governs regularities, while symmetry breaking brings texture. In artificial neural networks, symmetry has been a central design principle to efficiently capture regularities in the world, but the role of symmetry breaking is not well understood. Here, we develop a theoretical framework to study the *geometry of learning dynamics* in neural networks, and reveal a key mechanism of explicit symmetry breaking behind the efficiency and stability of modern neural networks. To build this understanding, we model the discrete learning dynamics of gradient descent using a continuous-time Lagrangian formulation, in which the learning rule corresponds to the kinetic energy and the loss function corresponds to the potential energy. Then, we identify *kinetic symmetry breaking* (KSB), the condition when the kinetic energy explicitly breaks the symmetry of the potential function. We generalize Noether's theorem known in physics to take into account KSB and derive the resulting motion of the Noether charge: *Noether's Learning Dynamics* (NLD). Finally, we apply NLD to neural networks with normalization layers and reveal how KSB introduces a mechanism of *implicit adaptive optimization*, establishing an analogy between learning dynamics induced by normalization layers and RMSProp. Overall, through the lens of Lagrangian mechanics, we have established a theoretical foundation to discover geometric design principles for the learning dynamics of neural networks.

With the rapid increase in available data and computational power, artificial neural networks have become an essential tool both in science and engineering. To obtain an accurate model, one has to overcome the challenge of tuning millions of randomly initialized parameters $q \in \mathbb{R}^N$ to minimize the loss function $f(q) \in \mathbb{R}$, a measure of the discrepancy between the model's predictions from the input data and the provided truth. In minimizing the loss, the learning rule specifies how to iteratively update these parameters based on the recent trajectory and the local geometry of the non-convex loss landscape. We can represent the repeated updates of the millions of parameters

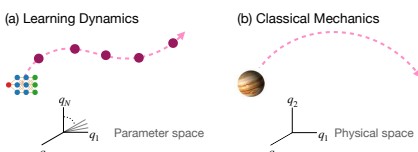

Figure 1: **Learning dynamics in analogy to classical mechanics.** (a) Learning dynamics of a neural network in parameter space. (b) Motion of a particle in physical space.

during learning as discrete movements of a point in high-dimensional parameter space (Fig. 1a). The architecture shapes the static loss landscape and the learning rule governs the motion of the model through parameter space during training. However, it is the complex interaction of the architecture and learning rule which determines the final state of the trained model. Inspired by tools and ideas from classical mechanics (Fig. 1b), we develop a theoretical framework which unifies the effects of the architecture and learning rule on the *geometry of the learning dynamics*.

---

\* Correspondence to: Hidenori Tanaka <hidenori.tanaka@ntt-research.com>

[2] Work done during an internship at Physics & Informatics Laboratories, NTT Research, Inc.

35th Conference on Neural Information Processing Systems (NeurIPS 2021).

**Symmetry as a mathematical lens.** One of the most powerful mathematical tools for studying the geometry of complex objects is symmetry. A symmetry of an object is a transformation that when applied to the object leaves certain properties unchanged. For example, consider how a circle can be rotated around its center without changing its overall appearance. In otherwords, a circle has a continuous rotational symmetry. In particular, the form of the symmetry transformation constrains the form of the object, the circle is the only 2D shape that has continuous rotational symmetry. Objects with symmetry are not restricted to geometric shapes, but could be functions or even dynamical systems. Recently, the lens of symmetry has been applied in the study of neural networks [1, 2]. A recent work [1] identified an array of symmetries in neural networks to study the geometry of the loss function $f(q)$ *in the parameter space*. For example, batch normalization [3] defined as $\text{BN}(qx) = (qx - \text{E}[qx])/\sqrt{\text{Var}[qx]}$ introduces a scale symmetry in the parameters immediately preceding the normalization layer, $\text{BN}((1+s)qx) = \text{BN}(qx)$ for any positive scalar $s \in \mathbb{R}^+$.

**Geometry of learning dynamics.** What does the symmetry of the loss function in parameter space imply about the symmetry of the learning dynamics? In classical mechanics, a symmetry of the potential function in space usually implies the *equivariance* of the trajectory, namely, a solution of the Newton's equation of motion (grey) remains a solution under the symmetry transformations (red) as shown in Fig. 2(a). This implies that the time evolution of the potential energy is *invariant* under symmetry transformations at any point in the dynamics. However, this intuition does not hold in modern neural networks. Consider the concrete example of a VGG11 model with a batch normalization layer after each convolutional filter. The scale symmetry of batch normalization implies the loss function is invariant to the scaling transformation on the convolutional filters $q$, $f((1+s)qx) = f(qx)$, however the learning dynamics are not. To demonstrate this we train two VGG11 models on Tiny ImageNet with

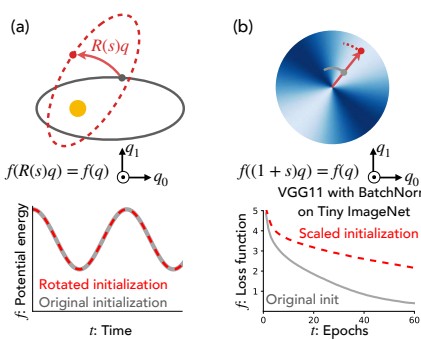

Figure 2: **Symmetry of the loss $\neq$ symmetry of the learning dynamics.** (a) Physical dynamics with rotational symmetry. (b) Learning dynamics in a parameter space with scale symmetry.

a standard initialization (grey) and a scaled initialization (red). Notice, that the scaling does not change the initial loss due to the scale symmetry, however it does effect the *learning dynamics* of the loss, as shown in the bottom of Fig. 2(b). In otherwords, even though the loss landscape observes scale symmetry at all steps in training, the learning dynamics do not. Understanding symmetry of the learning dynamics requires a new theoretical perspective.

**Implicit acceleration and variational perspective on learning dynamics.** In physics, variational formulations based on the Lagrangian and Hamiltonian provide a unifying framework to study the role of symmetry in dynamical systems. However, there is still a gap between the tools and concepts in physics and the learning dynamics of neural networks. We identify the root cause of this gap to be the learning rule, namely the fact that the continuous-time limit of gradient descent is a first order differential equation in time, gradient flow $\frac{dq}{dt} = -\nabla f$, unlike classical mechanics which is governed by Netwon's second law $m\frac{d^2q}{dt^2} = -\nabla f$, which is second-order in time. We bridge this gap by realizing that the effect of discretization, is to introduce an *implicit acceleration* term proportional to the learning rate. Leveraging this connection, we build the *mechanics of learning dynamics*:

**Sec. 1.** **From Newtonian to Lagrangian mechanics of learning via *implicit acceleration*.** We show how gradient descent with a finite learning rate introduces *implicit acceleration* making it amendable to the Lagrangian formulation of accelerated optimizers [4].

**Sec. 2.** **Unlike in physics, *kinetic symmetry breaking* (KSB) is inherent in learning dynamics.** We define *kinetic symmetry breaking*, where the kinetic energy corresponding to the learning rule explicitly breaks the symmetry of the potential energy corresponding to the loss function.

**Sec. 3.** **We derive *Noether's learning dynamics* (NLD) induced by the KSB.** We generalize Noether's theorem in physics to derive NLD, which accounts for damping, the unique symmetries of the loss, and the non-Euclidean metric used in learning rules.

**Sec. 4.** **KSB induces *impicit adaptive optimization* and enables successful learning.** We leverage NLD to exactly solve for the time-evolution of the effective learning rate for networks with normalization layers, and establish an exact analogy between two seemingly unrelated components of modern deep learning: normalization and adaptive optimization.

# 1   From Newtonian to Lagrangian mechanics of learning

The fundamental law of mechanics is the principle of least action [5]. We first review how pioneering works by Wibisono, Wilson, and Jordan [4, 6, 7] apply the variational principles of physics to model accelerated methods in optimization. We then show how gradient descent with a finite learning rate introduces *implicit acceleration* making it amendable to the Lagrangian formulation.

**Variational formulation of classical mechanics.** Just over a century after Newton, Joseph-Louis Lagrange demonstrated how Newton's equations of motion could be derived from a variational framework of least action. Rather than integrating an equation of motion over time to get the path an object takes from point $a$ to point $b$, Lagrange prescribed that the path taken by the object is the path that minimizes the action, the path integral of a function termed the Lagrangian, over all possible paths from $a$ to $b$. This principle of least action transformed mechanics from a differential description to a variational description. Consider for example the simple setup of a particle of mass $m$ at position $q \in \mathbb{R}^N$ in $N$ dimensional physical space moving according to a potential $f(q) \in \mathbb{R}$ and Newton's second law $m\ddot{q} = -\nabla f$, where $\ddot{q} = \frac{d^2 q}{dt^2}$ is the acceleration. Let the Lagrangian be the function $\mathcal{L}(q, \dot{q}, t) = \frac{1}{2}m\dot{q}^2 - f(q)$, the difference between the kinetic energy $T = \frac{1}{2}m\dot{q}^2$ and the potential energy $V = f(q)$ of the particle. The principle of least action states that the time evolution of the particle's position $q(t)$ is the function that minimizes the action $S[q] = \int_{t_0}^{t_1} \mathcal{L}(q, \dot{q}, t)dt$. In otherwords, $q(t)$, with boundary conditions $q(t_0) = a$ and $q(t_1) = b$, must satisfy the stationary condition $\frac{\delta S}{\delta q(t)} = 0$, which is equivalent to satisfying the Euler-Lagrange equation (see SI Sec. A for the review),

$$\frac{d}{dt}\left(\frac{\partial \mathcal{L}}{\partial \dot{q}}\right) = \frac{\partial \mathcal{L}}{\partial q}. \tag{1}$$

Indeed, substituting the Lagrangian into the Euler-Lagrange equation, we recover the Newton's equation of motion $m\ddot{q} = -\nabla f$ confirming the consistency of this variational formulation.

**Damping in optimization.** In learning dynamics, one of the universal features is damping, a term proportional to the velocity $\dot{q} = \frac{dq}{dt}$ in the equation motion. Intuitively, damping is needed for any optimization, since without damping the learning dynamics would conserve energy and not converge. We can incorporate damping into the previous Lagrangian example by introducing an explicit time-dependence $\mathcal{L}(q, \dot{q}, t) = e^{\frac{\mu}{m}t}(\frac{1}{2}m\dot{q}^2 - f(q))$, where $\mu$ is the damping (or friction) coefficient [8]. Substituting this Lagrangian into the Euler-Lagrange equation and dividing both sides by $e^{\frac{\mu}{m}t}$, we obtain the equation of motion of a damped particle $m\ddot{q} + \mu\dot{q} = -\nabla f$ as intended.

**Non-Euclidean geometry in optimization.** Many optimization algorithms assume an underlying geometry. For example, consider gradient descent with learning rate $\eta$. The update equation, $q(t + \eta) = q(t) - \eta\nabla f$, is equivalent to solving the minimization problem, $q(t + \eta) = \arg\min_{q'}\left[\langle\nabla f(q(t)), q'\rangle + \eta^{-1}\frac{1}{2}|q' - q(t)|^2\right]$, where $\langle\,,\,\rangle$ represents inner product. In otherwords, we can interpret each update of gradient descent as a trade-off (specified by $\eta$) between minimizing the loss $f(q)$, while remaining close to its current location in Euclidean distance. For non-Euclidean measures of distance we would obtain different optimization algorithms. Mirror descent generalizes this idea by replacing the distance measure by the Bergman divergence $D_h(y, x) = h(y) - h(x) - \langle\nabla h(x), y - x\rangle$ where $h(x)$ is a distance-generating function. Indeed, when the distance generating function is $h(x) = \frac{1}{2}|x|^2$, then the Bregman divergence simplifies to be the squared Euclidean distance $D_h(y, x) = \frac{1}{2}|x - y|^2$ and mirror descent is equivalent to gradient descent. The mirror descent framework encapsulates a broad array of optimization methods such as gradient descent, natural gradient descent, and others [9].

| | $\alpha_t$ | $\beta_t$ | $\gamma_t$ | $h(x)$ | Euler-Lagrange equation |
|---|---|---|---|---|---|
| Nesterov momentum | $\log 2 - \log t$ | $2\log t + \log 1/4$ | $2\log t$ | $\frac{1}{2}|x|^2$ | $\ddot{q} + 3/t + \nabla f = 0$ |
| Damped physical motion | $-\log m$ | $\log m$ | $\frac{\mu}{m}t$ | $\frac{1}{2}|x|^2$ | $m\ddot{q} + \mu\dot{q} + \nabla f = 0$ |
| Natural gradient flow | $-\log m$ | $\log m$ | $\frac{\mu}{m}t$ | $h(x)$ | $\dot{q} + F^{-1}\nabla f = 0 \ (m \to 0)$ |
| GD + momentum | $-\log\frac{\eta(1+\beta)}{2}$ | $\log\frac{\eta(1+\beta)}{2}$ | $\frac{2(1-\beta)}{\eta(1+\beta)}t$ | $\frac{1}{2}|x|^2$ | $\frac{\eta}{2}(1+\beta)\ddot{q} + (1-\beta)\dot{q} + \nabla f = 0$ |

Table 1: The Bregman Lagrangian [4, 6] provides a unified description of a family of learning rules with time dependent parameters $\alpha_t$, $\beta_t$, $\gamma_t$ and metric $h(x)$ which provide the algorithmic degrees of freedom.

**The Bregman Lagrangian.** Integrating building blocks of optimization such as damping and non-Euclidean geometry into the variational framework of mechanics, Wibisono, Wilson, and Jordan introduced the Bregman Lagrangian [4, 6],

$$\mathcal{L}(q, \dot{q}, t) = e^{\alpha_t + \gamma_t} \left( D_h(q + e^{-\alpha_t}\dot{q}, q) - e^{\beta_t} f(q) \right). \tag{2}$$

For various choices of the time-dependent factors $\alpha_t$, $\beta_t$, and $\gamma_t$ the Bregman Lagrangian, under the principle of least action, defines continuous approximations to an array of learning rules such as Nesterov's accelerated gradient methods [10, 11] and natural gradient flow [9]. The flexibility of this framework makes our analysis generalize to a broad array of learning rules, as summarized in table 1.

**Implicit acceleration and the Lagrangian formulation of gradient descent.** In the continuous-time limit of diminishing learning rate $\eta \to 0$, gradient descent $q(t + \eta) = q(t) - \eta\nabla f$ becomes a first order differential equation, gradient flow $\dot{q} = -\nabla f$. Unlike Nesterov's momentum [11], the trajectory of gradient descent with heavy ball momentum $\beta$ as applied in deep learning [12], follows gradient flow with the rescaling $(1 - \beta)\dot{q} = -\nabla f$. Here, we apply modified equation analysis [1, 13] to incorporate the effect of a finite learning rate and leverage the fact that the discrete steps of gradient descent closely follow the continuous-time trajectory of the second-order differential equation in time. We can informally see this by Taylor expanding an update with respect to a small, but finite learning rate $q(t + \eta) = q + \eta\dot{q} + \frac{\eta^2}{2}\ddot{q} + O(\eta^3)$ and then plugging this expression to the gradient descent updates $\frac{q(t+\eta)-q(t)}{\eta} = -\nabla f$. We keep the first order in $\eta$ and obtain a second-order differential equation $\frac{\eta}{2}\ddot{q} + \dot{q} = -\nabla f$. We call the additional term $\frac{\eta}{2}\ddot{q}$, *implicit acceleration*. This dynamics reduces to gradient flow $\dot{q} = -\nabla f$ in the continuous-time limit as expected. (Note that a complementary approach is to view the effects of discrete step size as *implicit gradient regularization* on the original loss $f(q) \to f(q) + \frac{\eta}{4}|\nabla f|^2$ [1, 14, 15, 16, 17].) Performing a similar analysis [1, 13] accounting for the effects of the heavy ball momentum with weight decay $k$ yields a second-order differential equation,

$$\frac{\eta}{2}(1 + \beta)\ddot{q} + (1 - \beta)\dot{q} + \nabla f(q) + kq = 0, \tag{3}$$

where $\frac{\eta}{2}(1 + \beta)\ddot{q}$ represents the implicit acceleration.

Indeed, we can connect this second-order dynamics to the Bregman Lagrangian by considering the factors $\alpha_t = -\log\frac{\eta(1+\beta)}{2}$, $\beta_t = \log\frac{\eta(1+\beta)}{2}$, $\gamma_t = \frac{2(1-\beta)}{\eta(1+\beta)}t$, and the distance generating function $h(x) = \frac{1}{2}|x|^2$. The Lagrangian describing the learning dynamics under gradient descent with a finite learning rate $\eta$, heavy ball momentum $\beta$, the loss function $f(q)$, and weight decay constant $k$ is

$$\mathcal{L}(q, \dot{q}, t) = e^{\frac{2(1-\beta)}{\eta(1+\beta)}t} \left[ \frac{\eta(1+\beta)}{4}|\dot{q}|^2 - \left( f(q) + \frac{k}{2}|q|^2 \right) \right]. \tag{4}$$

## 2 Symmetry and symmetry breaking in neural networks

The variational perspective of optimization, discussed in the previous section, provides a useful lens to characterizing the path a model takes through parameter space during training. To study the geometry of this path we will turn to the tools of symmetry. In particular, in this section we will discuss not only when symmetries exist in the loss function, but also when they are broken by the learning rule.

### 2.1 Symmetries of the loss function

There have been a number of works studying symmetries of the loss function in parameter space, such as [18, 19] who study discrete permutation symmetries or [1] who study continuous differentiable symmetries. In this work we will focus on continuous differentiable symmetries found in modern deep learning architectures. Consider a one-parameter family of differentiable maps $q \to Q(q, s)$ parameterized by a scalar $s \in \mathbb{R}$ such that $s = 0$ gives an identity $Q(q, 0) = q$. We say the potential $f(q)$ possesses a differentiable symmetry if its invariant to the transformation $f(Q(q, s)) = f(q)$ for all $s$. The symmetries we consider in this work are: *translation symmetry* where $Q(q, s) = q + s\hat{n}$ and $\hat{n}$ is a time-independent vector, *rotation symmetry* where $Q(q, s) = R(s)q$ and $R(s)$ is a rotation matrix, and *scale symmetry* where $Q(q, s) = (1 + s)q$. Translation symmetries occur in a neural network for parameters immediately preceding a softmax function. Rotation symmetry is one of the

most fundamental symmetry groups in classical mechanics and occur in neural networks with weight decay [20, 21] or word embeddings [22]. Scale symmetries appear ubiquitously in neural networks for parameters immediately preceding normalization layers, such as batch normalization [3] and its variants [23]. There is also a related symmetry, defined as rescale symmetry in [1], present in neural networks with continuous, homogeneous activation functions (e.g. ReLU, Leaky ReLU, linear).

## 2.2 Symmetry breaking in the kinetic energy

In the previous section, we introduced the Lagrangian formulation of learning dynamics, in which the learning rule corresponds to the kinetic energy and the loss function corresponds to the potential energy. The kinetic energy of the Bregman Lagrangian, defined as

$$T_h(q, \dot{q}, t) \equiv e^{\alpha_t} D_h(q + e^{-\alpha_t} \dot{q}, q) = e^{\alpha_t} \left( h(q + e^{-\alpha_t} \dot{q}) - h(q) \right) - \left\langle \nabla h(q), e^{-\alpha_t} \dot{q} \right\rangle \right), \quad (5)$$

is as important as the loss function, but its properties are less discussed. Here, we ask if the Bregman kinetic energy respects the same symmetries as the loss function, whether $T_h(Q(q, s), \dot{Q}(q, s), t) = T_h(q, \dot{q}, t)$.

**Euclidean kinetic energy.** First, we consider when $h(x) = \frac{1}{2}|x|^2$ and study the symmetry properties of the Euclidean kinetic energy,

$$T_{\mathrm{E}}(q, \dot{q}, t) = \frac{e^{-\alpha_t}}{2} |\dot{q}|^2. \quad (6)$$

For translation symmetry the velocity is invariant under the transformation $\dot{Q} = \dot{q}$ and thus indicating that the Euclidean kinetic energy is invariant under spatial translation. For rotation symmetry, the temporal differentiation commutes with the rotation $\dot{Q} = R\dot{q}$, however the Euclidean kinetic energy is invariant under spatial rotation because the norm is unchanged $|\dot{Q}|^2 = \langle R\dot{q}, R\dot{q} \rangle = |\dot{q}|^2$. However, for scale symmetry $\dot{Q} = (1 + s)\dot{q}$ and thus the norm $|\dot{Q}|^2 = (1 + s)^2 |\dot{q}|^2$ explicitly depends on $s$. Therefore, the kinetic energy explicitly breaks the scale symmetry present in the loss function. We call this explicit symmetry breaking of the Lagrangian by the kinetic energy term *Kinetic Symmetry Breaking* (KSB).

**Non-Euclidean kinetic energy.** Next, we consider the setting of a general non-Euclidean metric. The Taylor expansion of the Bregman kinetic energy with respect to the first term $h(q + e^{-\alpha_t} \dot{q})$ in Eq. 5 results in the simplification,

$$T_h(q, \dot{q}, t) = (e^{-\alpha_t}/2) \left\langle \dot{q}, \nabla^2 h(q)\dot{q} \right\rangle + O\left( (e^{-\alpha_t})^2 \right). \quad (7)$$

Notice, the Euclidean kinetic energy only depends on the squared norm of the velocity $|\dot{Q}|^2$ because the Hessian is the identity matrix $\nabla^2 \frac{1}{2}|x|^2 = I$ and the higher order derivatives are zero. With non-Euclidean metrics, such simplifications special to the Euclidean case do not necessarily apply and we should not expect that the Bregman kinetic energy respects any symmetries in general. However, the natural gradient [9] and its elegant approximations, such as [24], are examples of non-Euclidean optimization methods, which have invariance properties by construction. While the invariance properties of the natural gradient descent can be broken due to discretization [25], the fast and accurate approximation of natural gradients in modern large-scale neural networks is an active area of research [26, 27].

## 3 Noether's learning dynamics

So far, we have seen how the Bregman Lagrangian provides a unified variational description of learning dynamics, and the scale symmetry, ubiquitous in modern neural networks, induces kinetic symmetry breaking (KSB). Here, we apply Noether's theorem to the Bregman Lagrangian and derive *Noether's Learning Dynamics* (NLD), a unified equality that holds for any combination of symmetry and learning rules.

**Noether's theorem.** In 1918 Emmy Noether famously described one of the most fundamental theorems in modern physics: every differentiable symmetry of the action has an associated conserved quantity through time [28]. Noether's theorem not only proves the existence of, but also provides an explicit expression for, the conserved quantity, often called Noether charge. Focusing just on the symmetry properties of the Lagrangian, a general form of Noether's theorem relates the dynamics of

the Noether charge $\langle \partial_{\dot{q}} \mathcal{L}, \partial_s Q \rangle$ to the change in the Lagrangian under an infinitesimal transformation $\partial_s \mathcal{L}$ as

$$\frac{d}{dt} \langle \partial_{\dot{q}} \mathcal{L}, \partial_s Q \rangle = \partial_s \mathcal{L}, \tag{8}$$

where we evaluate all derivatives at the identity element of the symmetry group ($s = 0$). We can derive this relationship as $\partial_s \mathcal{L} = \langle \partial_q \mathcal{L}, \partial_s Q \rangle + \langle \partial_{\dot{q}} \mathcal{L}, \partial_s \dot{Q} \rangle = \frac{d}{dt} \langle \partial_{\dot{q}} \mathcal{L}, \partial_s Q \rangle$, where we plugged in the Euler-Lagrange equation $\partial_q \mathcal{L} = \frac{d}{dt}(\partial_{\dot{q}} \mathcal{L})$ and rearranged the total time derivative. The most common application of Noether's theorem in physics is when the Lagrangian, including both kinetic and potential energies, is symmetric under the infinitesimal transformation, $\partial_s \mathcal{L} = 0$, implying Noether charge is conserved through time. For example, if we assume symmetry of the Lagrangian due to homogeneity or isotropy of space, Noether's theorem directly implies the conservation of momentum or angular momentum respectively. However, as shown in Sec. 2, the Bregman kinetic energy for learning systems is often asymmetric to the symmetry of the potential function, implying that the resulting Noether charge is *not* conserved.

**Noether's learning dynamics.** Here, we derive a general equation describing the kinetic-asymmetry-induced dynamics for the Noether charge in learning systems. If the Bregman kinetic energy does not respect the symmetry of the loss function, then Eq. 8 becomes

$$\frac{d}{dt} \langle \partial_{\dot{q}} \mathcal{L}, \partial_s Q \rangle = e^{\gamma_t} \partial_s T_h, \tag{9}$$

implying kinetic symmetry breaking ($\partial_s T_h \neq 0$) induces motion of the Noether charge. To build better intuition for this motion and inspired by the definition of the generalized momentum, we define the *Bregman momentum* as,

$$\Delta_h \equiv e^{-\gamma_t} \partial_{\dot{q}} \mathcal{L} = \nabla h(q + e^{-\alpha_t} \dot{q}) - \nabla h(q). \tag{10}$$

Expanding Eq. 9 by plugging in the definition of the Bregman Lagrangian Eq. 2, the Bregman kinetic energy Eq. 5, and the Bregman momentum Eq 24 (details are described in SI Sec. B), we obtain *Noether's Learning Dynamics* (NLD),

$$\overbrace{\frac{d}{dt} \langle \Delta_h, \partial_s Q \rangle}^{\text{charge}} + \overbrace{\dot{\gamma}_t \langle \Delta_h, \partial_s Q \rangle}^{\text{damping}} = \overbrace{\langle \Delta_h, \partial_s \dot{Q} \rangle}^{\text{kinetic asymmetry}} + \overbrace{e^{\alpha_t} \langle \Delta_h - e^{-\alpha_t} \nabla^2 h(q) \dot{q}, \partial_s Q \rangle}^{\text{non-Euclidean metric}}. \tag{11}$$

Each term in NLD has an intuitive meaning. Recall the conventional setting in classical mechanics, where the entire Lagrangian is symmetric, there is no damping, and the metric is Euclidean. In this case, the Noether charge is the inner product of momentum and the generator of the symmetry transformation $\langle e^{-\alpha_t} \dot{q}, \partial_s Q \rangle$. The first term of NLD, $\langle \Delta_h, \partial_s Q \rangle$, is the inner product of the Bregman momentum and the generator of the symmetry transformation. In the setting of no damping, this terms is a generalization of the conventional Noether charge that encompasses non-Euclidean metrics. Indeed, in the Euclidean case, we can see that $\Delta_E = e^{-\alpha_t} \dot{q}$ and the term reduces to the conventional Noether charge as expected. The second term of NLD, $\dot{\gamma}_t \langle \Delta_h, \partial_s Q \rangle$, represents the contribution of damping and the term becomes zero when $\gamma_t$ is constant and there is no damping. The third term of NLD, $\langle \Delta_h, \partial_s \dot{Q} \rangle$, represents the contribution of kinetic symmetry breaking. Under the Euclidean metric, this term is proportional to $\langle \dot{q}, \partial_s \dot{Q} \rangle$. For translation symmetry, $\partial_s \dot{Q} = 0$ and this term is zero $\langle \Delta_h, 0 \rangle = 0$. Similarly, for rotation symmetry, this term is zero $\langle R|_{s=0} \dot{q}, (\partial_s R) \dot{q} \rangle = \langle \dot{q}, (\partial_s R) \dot{q} \rangle = 0$. Here, we used the fact that the generator of rotation is a skew-symmetric matrix. However, for scale symmetry, $\partial_s \dot{Q} = \dot{q}$ unique to learning systems, this term is proportional to $|\dot{q}|^2$ and thus present. The fourth term of NLD, $e^{\alpha_t} \langle \Delta_h - e^{-\alpha_t} \nabla^2 h(q) \dot{q}, \partial_s Q \rangle$, represents the contribution of the non-Euclidean geometry and is zero under the Euclidean metric $h(x) = \frac{1}{2}|x|^2$.

**Noether's learning dynamics under gradient flow.** Next, we show that in the over-damped limit of NLD, the charge $\langle \Delta_h, \partial_s Q \rangle$ itself, rather than its time-derivative becomes zero. Consider Eq. 11 when $\alpha_t = -\log m$, $\beta_t = \log m$, $\gamma_t = \frac{\mu}{m} t$,

$$m \frac{d}{dt} \langle \Delta_h, \partial_s Q \rangle + \mu \langle \Delta_h, \partial_s Q \rangle = m \partial_s T. \tag{12}$$

When $\partial_s T = 0$, this equation describes how the charge decays exponentially with damping $\langle \Delta_h, \partial_s Q \rangle = e^{-\frac{\mu}{m} t}$ as in Fig. 3. In the mass-less limit $m \to 0$ with a fixed friction $\mu$, the terms proportional to $m$ in Eq. 12 all vanish and we get

$$\langle \Delta_h, \partial_s Q \rangle = 0, \tag{13}$$

which states that the charge itself is zero, rather than its time derivative. Furthermore, in the Euclidean settings, the result simplifies to $\langle \dot{q}, \partial_s Q \rangle = 0$. As a physical analogy, this means that quantities such as momentum or angular momentum becomes zero in the over-damped limit. This formula unifies an array of conservation properties noticed under gradient flow [1, 29, 30, 31, 32] with a formal theoretical connection to Noether's theorem.

Overall, just like how Noether's theorem [28] unified an array of conservation laws and provided a theoretical foundation to discover new ones in physics, we have further unified conservation laws previously observed in learning systems [1, 29, 30, 31, 32] and generalized these results for any combination of differentiable symmetries in neural network architectures and learning rules (e.g., natural gradient descent, Nesterov's accelerated gradient descent, accelerated mirror descent, and cubic-regularized Newton's method).

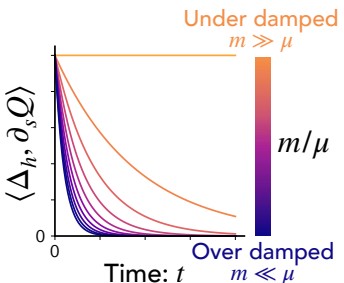

Figure 3: **Noether's theorem with damping.** Without KSB, the charge decays exponentially, but in the underdamped limit $m/\mu \gg 1$, the charge becomes a conserved quantity. In the overdamped limit $m \to 0$, the charge exponentially diminishes to zero.

## 4 Kinetic symmetry breaking induces implicit adaptive optimization

In the previous sections, we identified the mechanism of kinetic symmetry breaking and derived the Noether's learning dynamics (NLD). Here, we explore a role of symmetry breaking in neural networks by applying NLD to study auto rate-tuning by normalization layers, originally studied by Arora, Li, and Lyu [33]. In particular, we establish an exact theoretical analogy between two seemingly unrelated key components of modern deep learning: normalization and adaptive optimization.

Consider a neural network model with normalization layers (such as batch normalization [3], group normalization [34], layer normalization [35], and others [23]), which introduces scale symmetry in the potential (loss) function $f((1 + s)q) = f(q)$. We train this model with gradient descent with a finite learning rate $\eta$, momentum $\beta$, and weight decay $k$. We are interested in the evolution of the potential function through the learning dynamics $f(q(t))$. Due to the scale symmetry of the network introduced by the normalization layer, the time evolution on the loss landscape only depends on the directional vector $\hat{q} \equiv q/|q|$ of the parameters throughout training, $f(q(t)) = f(\hat{q}(t))$.

What is the equation of motion for the directional vector? To solve for the dynamics, we first move to a polar coordinate system $q = r\hat{q}$ where $r = |q|$ is the radial norm and $\hat{q}$ is the directional vector. This is akin to how we solve for the motion of a planet in a central field in classical mechanics. From Eq. 4, the Lagrangian for this learning system in polar coordinates is,

$$\mathcal{L}_c = e^{\frac{\mu}{m}t}\left[\frac{m}{2}\dot{r}^2 + \left(\frac{m}{2}\left|\dot{\hat{q}}\right|^2 - \frac{k}{2}\right)r^2 - f(\hat{q})\right] + \lambda(t)\left(|\hat{q}|^2 - 1\right), \tag{14}$$

where $m = \frac{\eta(1+\beta)}{2}$, $\mu = 1 - \beta$, and $C(\hat{q}) = |\hat{q}|^2 - 1 = 0$ is a Lagrange multiplier enforcing the unit norm constraint. The Lagrangian formulation facilitates coordinate transformations, because the form of the Euler-Lagrange equation is invariant under coordinate transformations. Taking advantage of this fact, we apply the Euler-Lagrange equation to derive the equation of motion for the directional vector as (see SI Sec. C for the derivation)

$$m\ddot{\hat{q}} + \mu\dot{\hat{q}} = -\frac{1}{r^2(t)}\nabla f(\hat{q}), \tag{15}$$

where $\nabla f(\hat{q})$ represents the gradient of $f$ evaluated at $\hat{q}$. Notice the striking similarity of this equation of motion with the equation of motion of the original parameters $m\ddot{q} + \mu\dot{q} = -\nabla f(q)$. This implies that the time evolution of the loss for a network with normalization layers trained by gradient descent is equivalent to the time evolution of the loss for an identical network with unit filter norm trained by an adaptive optimizer which scales the gradients by a factor of $r^2(t)$.

Strikingly, NLD describes the motion of this adaptive scaling factor $r^2(t)$. When $\alpha_t = -\log m$, and $h(x) = \frac{1}{2}|x|^2$, then the charge is $\langle \Delta_h, \partial_s Q \rangle = m\langle \dot{q}, q \rangle = \frac{1}{2}m\partial_t r^2$ and Eq. 11, including the effect

of weight decay $k$, implies

$$m\partial_t^2 r^2 + \mu\partial_t r^2 = -2kr^2 + \frac{2m}{\mu^2 r^2}|\nabla f(\hat{q})|^2, \qquad (16)$$

where we kept the first order terms in $m$ and $k$. In the over-damped regime, ignoring inertia, we get a change in the squared parameter norm $r^2(t)$ as (see SI Sec. D for the derivation):

$$r^2(t) = \sqrt{\frac{2\eta(1+\beta)}{(1-\beta)^3}\int_0^t e^{-\frac{4k}{1-\beta}(t-\tau)}|\nabla f(\hat{q}(\tau))|^2 d\tau + e^{-\frac{4k}{1-\beta}t}r^4(0)}. \qquad (17)$$

Each term in this implicit adaptive learning rate has an intuitive meaning. The first integral term keeps track of the recent history of the magnitude of gradient norms $|\nabla f(\hat{q}(\tau))|^2$ that the filters with fixed unit norm receive. The weight decay $k$ controls the time-scale of the short-term memory of the accumulated gradient norms through the exponential kernel $e^{-\frac{4k}{1-\beta}(t-\tau)}$. When $k = 0$ all the gradients will be accumulated resembling an AdaGrad optimizer [36]. Notably, this integral term is proportional to the learning rate $\eta$ implying that this mechanism of adaptive optimization is only present with a finite learning rate. The second term represents an exponentially decaying memory of initialization. The finite filter norm at initialization $r^4(0)$ prevents instability of learning dynamics avoiding division by zero.

So how does this adaptive scaling of the gradient step sizes compare with hand-designed adaptive optimizers successful in deep learning [36, 37, 38]? Here, we develop a continuous-time model of the explicit adaptive optimizer RMSProp and discover that the functional form of normalization layers' effective learning rate schedule Eq. 17 functionally agrees with that of RMSProp. The RMSProp algorithm is a recursive update rule expressed as $q_{n+1} = q_n - \frac{\eta}{\sqrt{G_n}}\nabla f(q_n)$, where the gradient is scaled by a factor of $\sqrt{G_n}$. The factor $G_n$ keeps track of the history of the gradient norms following $G_{n+1} = \rho G_n + (1-\rho)|\nabla f(q_n)|^2$, where $\rho$ is a hyperparameter. Solving the continuous-time dynamics $\eta\frac{dG}{dt} = -(1-\rho)G(t) + (1-\rho)|\nabla f(q(\tau))|^2$ of the adaptive scaling factor of RMSProp, we obtain

$$\sqrt{G(t)} = \sqrt{\frac{1-\rho}{\eta}\int_0^t e^{-\frac{1-\rho}{\eta}(t-\tau)}|\nabla f(q(\tau))|^2 d\tau + e^{-\frac{1-\rho}{\eta}t}G(0)}. \qquad (18)$$

Strikingly, we find that this functional form induced by RMSProp exactly matches with the implicit adaptive learning rate schedule Eq. 17 due to normalization layers suggesting potential benefits of the implicit adaptive optimization induced by normalization.

In summary, our theory has yielded the following new insights:

1. The optimization geometry of SGD with a finite step size $\eta$, momentum $\beta$, and weight decay $k$ together breaks the scale symmetry of the loss (e.g., with normalization layers) and generates an implicit mechanism of adaptive optimization akin to RMSProp.

2. The implicit adaptive optimization mechanism exists when the learning dynamics is in the underdamped regime $m/\mu \propto \eta/(1-\beta) \gg 0$ and thus requires the learning rate to be finite.

3. Momentum significantly amplifies the effect of adaptive optimization by $\frac{1+\beta}{(1-\beta)^3} = 1900$ folds as in Eq. 17 even in the standard setting of $\beta = 0.9$.

4. For scale-invariant parameters, symmetry breaking due to weight decay $1-k$ plays a role similar to the discount factor $\rho$ for the cumulative gradient norm in RMSProp.

Finally, we empirically validate these predictions by training convolutional neural networks with batch normalization (VGG11) on a large data set (Tiny-ImageNet) under various hyperparameter settings as in Fig. 4 (see SI Sec. E). In addition to the standard training with an "unconstrained filter norm" (pink), we perform ablation experiments with a "constrained filter norm" (blue). As a concrete example, consider a convolutional filter $q \in \mathbb{R}^n$ preceding the batch normalization layer. The loss is invariant under the scaling transformation of the filter norm $f((1+s)q) = f(q)$. To test the presence and benefits of implicit adaptive optimization to the learning dynamics of the loss, we freeze the dynamics of the filter norm by repeatedly performing the operation $q(t) \leftarrow \frac{q(t)}{|q(t)|}|q(0)|$ after each step of optimization [33]. This enforces the filter norm to be fixed $|q(t)| = |q(0)|$ throughout training for any $t$. When trained with a constant small learning rate $\eta \sim 0.001$, the final test accuracy (A-1)

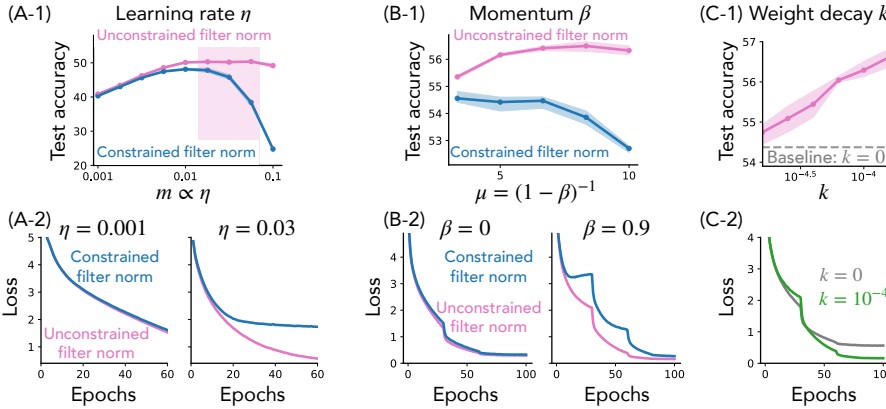

Figure 4: **Broken-symmetry induced dynamics of the filter norm is a key to successful learning.** (VGG-11 models trained on Tiny-ImageNet.) (A-1) The final test accuracy and (A-2) the loss dynamics of models trained at various constant learning rates $\eta$ with $\beta = 0.9$. (B-1) The final test accuracy and (B-2) the loss dynamics of models trained with various momentum $\beta$ with learning rate drops. In agreement with our theory, the advantage of implicit adaptive optimization is present when trained with large learning rates and large momentum. (C-1) The final test accuracy and (C-2) the loss dynamics when the model is trained with various weight decay rates $k$, validating its benefits.

and loss dynamics (A-2) of models with unconstrained filter norm (pink) and constrained filter norm (blue) are almost identical. However, in a large learning rate regime $\eta \sim 0.03$, the models trained with unconstrained filter norm (pink) outperforms the models trained with constrained filter norm (blue) demonstrating the existence and benefits of implicit adaptive optimization. Similarly, we confirm our prediction that the presence of momentum $\beta \sim 0.9$ is essential to amplify the effects of implicit adaptive optimization, as seen in test accuracy (B-1) and loss dynamics (B-2). Finally, we demonstrate the benefits of symmetry breaking due to weight decay $k$ which acts in analogy with the discounting factor $\rho$ for the cumulative gradient norms of RMSProp. Indeed, the final test accuracy (C-1) and the learning dynamics of the loss (C-2) are both enhanced by the presence of weight decay $k$ in agreement with [39]. Overall, the hyperparameters of the modern optimization - learning rate, momentum, and weight decay - each play an important role in breaking the scale symmetry of the loss function and providing an mechanism of implicit adaptive optimization, a key to successful learning in neural networks.

# 5 Related works and future directions

**Finite learning rates, modified gradient flow, and generalization.** Recently, many works have demonstrated that an initial large learning rate is a key to successful training [40, 41, 42, 43, 44]. Another line of works [1, 13, 14, 15, 16, 45, 46] have incorporated the effect of a finite learning rate into gradient flow by adding a modification term. A common approach applies modified equation analysis to replace the original loss function $f$ by a modified loss function $\tilde{f} = f + \frac{\eta}{4}|\nabla f|^2$ [1, 14, 15], where the additional term is called *implicit gradient regularization*. Intuitively, this term biases the learning dynamics towards wider minima with smaller gradient norms. To test this intuition empirically, recent works have explicitly added the gradient regularization term $\frac{\eta}{4}|\nabla f|^2$ to the original loss and have shown that the models trained with the modified loss generalize well even with small learning rates [16] or full-batch [17]. In this work, we have taken a complementary approach and applied modified equation analysis *in time* to add an *implicit acceleration* term $\frac{\eta}{2}\ddot{q}$ [1, 13, 45]. When training with a finite learning rate, this implicit acceleration term gives rise to the implicit adaptation of the gradients. Remarkably, the two complementary pictures (i.e., implicit gradient regularization and implicit acceleration) offer a consistent view on how the large learning rates provide mechanisms to properly regularize the norm of gradients to avoid training instabilities. An idea for future work is to directly simulate gradient flow with implicit acceleration, instead of gradient flow with [15, 16, 17] or without [47] implicit gradient regularization, to keep the benefits of finite learning rates while avoiding the computations of the Hessian gradient products.

**Geometry of optimization.** The geometric property of optimization has a long history of study. For example, natural gradient by Amari [9] and its approximations [24, 48] are invariant under affine transformations including scale transformation due to normalization layers [39]. With the rise of deep learning, there has been recent developments to efficiently approximate and apply natural gradients to large neural networks [24, 26, 48, 49, 50, 51, 52, 53]. When training deep neural networks, we still commonly use gradient-based optimization methods represented by stochastic gradient descent with momentum [12]. An array of adaptive optimization methods [36, 37, 38] have been proposed partially motivated as diagonal approximations of the Fisher Information Matrix (see recent discussions questioning the validity of the approximations [54]). A pioneering work [55] by Neyshabur et al. has identified that gradient descent does not respect the rescale symmetry introduced by ReLU activations, followed by works proposing to solve such discrepancies by enforcing symmetry in the learning dynamics [56, 57]. While above works have always tried to remove such broken symmetries, we demonstrated theoretically and empirically that the broken-symmetry can play an important role in deep learning, thus providing a unique counterexample. A future direction is to extend our analysis to the case of rescale symmetry of the homogemeous activation functions such as ReLU.

**Conservation laws in neural networks.** Balancing properties of network parameters have been identified in gradient flow dynamics of neural networks with linear [29] and ReLU [30, 31] activation functions. These properties have been leveraged to better understand the role of over-parameterization [29], implicit regularization [30], network pruning [31], continual learning [58], and self-supervised learning [32]. A recent work identified symmetry-induced geometric structures of the loss and studied their roles under gradient flow dynamics with modified loss predicting the parameter dynamics [1]. However, despite the ubiquity of conservation laws in deep learning, no formal connection to the Noether's theorem has been made due to the previous focus on the gradient flow dynamics. Here, we took an alternative path starting from a Lagrangian formulation, which encapsulates gradient flow as the mass-less limit. From this perspective, we unified all the conservation laws under the umbrella of Noether's theorem, and provided its generalization for accelerated and non-Euclidean methods in optimization. In future works, we can harness the generality of our theory to investigate the dynamics of more complex learning rules such as natural gradient descent with or without Nesterov's momentum in combination with any differentiable symmetries.

**Auto-rate tuning by normalization layers.** Batch normalization [3] and its variants [35, 59] are essential to train *deep* neural network models [60] providing multiple benefits as elegantly reviewed in [61]. An important benefit of normalization layers is to enable stable training with large learning rates [62]. A pioneering work [33] rigorously studied the benefits of monotonically decreasing the effective learning rate in simple setting without momentum nor weight decay. Another line of works noticed the role of weight decay to effectively enforce a higher learning rate [39, 63, 64] inspiring an automatic learning rate scheduler [65]. Li and Arora [66] theoretically studied the setting with weight decay and momentum to design a exponential learning rate schedule that still leads to successful training counter-intuitively. In the present analysis, we leveraged the Lagrangian formulation to handle the complexity of modern practical optimizers and derived an exact solution describing the time-evolution of the effective learning rate in the presence of momentum and weight decay. By developing a continuous-time description of adaptive optimizers in parallel, we discovered that the functional form of the effective learning rate exactly matches with that of RMSProp, where momentum significantly amplifies the effect, and weight decay acts in exact analogy with the discounting factor. In the context of thorough empirical studies on adaptive optimizers [67, 68, 69], our results raise the hypothesis that the benefit of an adaptive optimizers may not be as significant in networks with normalization layers as long as the hyper-parameters, in particular momentum, are properly tuned.

## 6 Conclusion.

In nature, symmetry governs regularities, while symmetry breaking brings texture [70], examples of which include superconductivity [71], nucleon mass [72], ring attractor neural networks [73, 74, 75], and the Higgs boson [76]. Despite many attempts to bring symmetry to neural networks [2, 55], symmetry breaking has rarely been discussed as a design principle [22], or even treated as an obstacle to successful learning [55, 56, 57]. In this work, we have discovered a novel role of explicit symmetry breaking in learning systems by applying a Lagrangian formulation to neural networks. Overall, understanding not only when symmetries exist, but how they are broken is essential to discover geometric design principles in neural networks.

## Acknowledgments and Disclosure of Funding

We thank Max Aalto, Kyosuke Adachi, Yuto Ashida, Fatih Dinc, Surya Ganguli, Kyogo Kawaguchi, Ekdeep Singh Lubana, Akiko Tanaka, Hajime Tanaka, Atsushi Yamamura, and Daniel Yamins for helpful discussions. D.K. thanks the Stanford Data Science Scholars program and NTT Research for support.

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
