# Supplementary Information

## A    The principle of least action and the Euler-Lagrange equation

Here, we review the principle of least action and the derivation of the Euler-Lagrange equation [5]. The principle of least (stationary) action states that the time evolution of the position of a particle $q(t) \in \mathbb{R}^N$ with boundary conditions ($q(t_0) = a$ and $q(t_1) = b$) is the function that minimizes the action,

$$S[q] = \int_{t_0}^{t_1} \mathcal{L}(q, \dot{q}, t) dt.$$

Now, let us derive the differential equation that gives a solution to the variational problem. Assuming $q$ is the solution, then it implies that any small variation $q \rightarrow q + \delta q$ will increase the value of the action. Thus, the stationary condition for the solution is,

$$\delta S = \int_{t_1}^{t_2} \mathcal{L}(q + \delta q, \dot{q} + \delta \dot{q}, t) dt - \int_{t_1}^{t_2} \mathcal{L}(q, \dot{q}, t) dt = \int_{t_1}^{t_2} \left( \frac{\partial \mathcal{L}}{\partial q} \delta q + \frac{\partial \mathcal{L}}{\partial \dot{q}} \delta \dot{q} \right) dt = 0. \quad (19)$$

By integrating the second term by parts, we get

$$\delta S = \left[ \frac{\partial \mathcal{L}}{\partial \dot{q}} \delta q \right]_{t_1}^{t_2} + \int_{t_1}^{t_2} \left( \frac{\partial \mathcal{L}}{\partial q} - \frac{d}{dt} \frac{\partial \mathcal{L}}{\partial \dot{q}} \right) \delta q \, dt = 0. \quad (20)$$

The first term is zero due to the fixed boundary conditions, $\delta q(t_1) = \delta q(t_2) = 0$. Thus, the second term must vanish for arbitrary variation $\delta q$. This condition yields the Euler-Lagrange equation,

$$\frac{d}{dt} \frac{\partial \mathcal{L}}{\partial \dot{q}} = \frac{\partial \mathcal{L}}{\partial q}. \quad (21)$$

## B    The derivation of the Noether's learning dynamics

**Noether's learning dynamics.**    *Consider the learning dynamics of a parameter vector $q \in \mathbb{R}^N$ governed by the Bregman Lagrangian $\mathcal{L}(q, \dot{q}, t) = e^{\gamma_t} \left( e^{\alpha_t} D_h(q + e^{-\alpha_t} \dot{q}, q) - e^{\alpha_t + \beta_t} f(q) \right)$. If the loss function $f(q)$ is invariant under a one-parameter family of differentiable maps $q(t) \rightarrow Q(q(t), s)$, i.e., $f(Q(q(t), s)) = f(q)$ for all $(q, s)$, then the learning dynamics obeys the following equality,*

$$\frac{d}{dt} \left\langle \Delta_h, \frac{\partial Q}{\partial s} \right\rangle + \dot{\gamma}_t \left\langle \Delta_h, \frac{\partial Q}{\partial s} \right\rangle = \left\langle \Delta_h, \frac{\partial \dot{Q}}{\partial s} \right\rangle + e^{\alpha_t} \left\langle \Delta_h - e^{-\alpha_t} \nabla^2 h(q) \dot{q}, \frac{\partial Q}{\partial s} \right\rangle, \quad (22)$$

*where $\Delta_h$ is the Bregman momentum defined as $\Delta_h \equiv e^{-\gamma_t} \partial_{\dot{q}} \mathcal{L} = \nabla h(q + e^{-\alpha_t} \dot{q}) - \nabla h(q)$.*

*Derivation.*    Here, we derive the Noether's learning dynamics by applying Noether's theorem to the Bregman Lagrangian. A general form of the Noether's theorem relates the dynamics of Noether charge $\frac{d}{dt} \left\langle \frac{\partial \mathcal{L}}{\partial \dot{q}}, \frac{\partial Q}{\partial s} \right\rangle$ to the change of Lagrangian at infinitesimal symmetry transformation $\frac{\partial \mathcal{L}}{\partial s}|_{s=0}$ evaluated at the identity transformation as,

$$\frac{\partial \mathcal{L}}{\partial s} = \left\langle \frac{\partial \mathcal{L}}{\partial q}, \frac{\partial Q}{\partial s} \right\rangle + \left\langle \frac{\partial \mathcal{L}}{\partial \dot{q}}, \frac{\partial \dot{Q}}{\partial s} \right\rangle = \frac{d}{dt} \left\langle \frac{\partial \mathcal{L}}{\partial \dot{q}}, \frac{\partial Q}{\partial s} \right\rangle,$$

where we plugged in the Euler-Lagrange equation $\frac{\partial \mathcal{L}}{\partial q} = \frac{d}{dt} \left\langle \frac{\partial \mathcal{L}}{\partial \dot{q}} \right\rangle$ and performed integration by parts. Since the loss function $f(q)$ is invariant under the differentiable transformation $q(t) \rightarrow Q(q(t), s)$, we have $\frac{\partial}{\partial s} f(Q) = 0$. Thus, we evaluate the dynamics of the Noether charge driven by the kinetic symmetry breaking,

$$\frac{d}{dt} \left\langle \frac{\partial \mathcal{L}}{\partial \dot{q}}, \frac{\partial Q}{\partial s} \right\rangle = e^{\gamma_t} \frac{\partial T_h}{\partial s}, \quad (23)$$

where $T_h$ is the kinetic energy of learning defined as,

$$T_h(q, \dot{q}, t) = e^{\alpha_t} \left( h(q + e^{-\alpha_t} \dot{q}) - h(q) - \langle \nabla h(q), e^{-\alpha_t} \dot{q} \rangle \right).$$

For immediate use, we first evaluate the derivative of the kinetic energy $T_h$ with respect to $q$ and $\dot{q}$ as,

$$\frac{\partial T_h}{\partial q} = e^{\alpha_t} \left( \nabla h(q + e^{-\alpha_t} \dot{q}) - \nabla h(q) - \langle \nabla^2 h(q), e^{-\alpha_t} \dot{q} \rangle \right) = e^{\alpha_t} \left( \Delta_h - \langle \nabla^2 h(q), e^{-\alpha_t} \dot{q} \rangle \right),$$

and

$$\frac{\partial T_h}{\partial \dot{q}} = \nabla h(q + e^{-\alpha_t} \dot{q}) - \nabla h(q) = \Delta_h. \tag{24}$$

By evaluating the left hand side of Eq. 23, we get

$$\frac{d}{dt} \left\langle \frac{\partial \mathcal{L}}{\partial \dot{q}}, \frac{\partial Q}{\partial s} \right\rangle = \frac{d}{dt} \left( e^{\gamma_t} \left\langle \frac{\partial T_h}{\partial \dot{q}}, \frac{\partial Q}{\partial s} \right\rangle \right) = \frac{d}{dt} \left( e^{\gamma_t} \left\langle \Delta_h, \frac{\partial Q}{\partial s} \right\rangle \right) = e^{\gamma_t} \left( \dot{\gamma}_t \left\langle \Delta_h, \frac{\partial Q}{\partial s} \right\rangle + \frac{d}{dt} \left\langle \Delta_h, \frac{\partial Q}{\partial s} \right\rangle \right).$$

By evaluating the right hand side of Eq. 23, we get

$$e^{\gamma_t} \frac{\partial T_h}{\partial s} = e^{\gamma_t} \left( \left\langle \frac{\partial T_h}{\partial q}, \frac{\partial Q}{\partial s} \right\rangle + \left\langle \frac{\partial T_h}{\partial \dot{q}}, \frac{\partial \dot{Q}}{\partial s} \right\rangle \right) = e^{\gamma_t} \left[ e^{\alpha_t} \left\langle \Delta_h - \nabla^2 h(q) e^{-\alpha_t} \dot{q}, \frac{\partial Q}{\partial s} \right\rangle + \left\langle \Delta_h, \frac{\partial \dot{Q}}{\partial s} \right\rangle \right].$$

Finally, by equating the left and right sides, and then dividing both sides by $e^{\gamma_t}$, we obtain the Noether's learning dynamics. $\qquad \square$

## C  Motion in scale-symmetric potential

In this section, we consider motion of a particle in scale-symmetric potential governed by the Lagrangian,

$$\mathcal{L}(q, \dot{q}, t) = e^{\frac{\mu}{m} t} \left( \frac{m}{2} |\dot{q}|^2 - \left( f(q) + \frac{k}{2} |q|^2 \right) \right), \tag{25}$$

where $q \in \mathbb{R}^N$ is a vector representing the general coordinate, $m$ is the mass, $\mu$ is the damping coefficient, $f(q)$ is the potential function with scale symmetry, and $k$ is the spring constant for the quadratic potential function. With the parameterization of $m = \frac{\eta}{2}(1 + \beta)$ and $\mu = 1 - \beta$, this dynamics is equivalent to the learning dynamics of the scale-invariant parameter vector $q$ under gradient descent with a finite learning rate $\eta$, momentum $\beta$, and weight decay $k$.

### C.1  Equations of motion

First, we leverage the scale symmetry of the potential (loss) function by separating the coordinate as $q = r\hat{q}$, where $r$ is the radial length of the parameter vector $q$, and $\hat{q}$ is the unit vector representing the direction. The time derivative is $\dot{q} = \dot{r}\hat{q} + r\dot{\hat{q}}$, and the Lagrangian can be re-written with this new coordinate as,

$$\mathcal{L} = e^{\frac{\mu}{m} t} \left( \frac{m}{2} \left( \dot{r}^2 + 2r\dot{r} \overbrace{\langle \hat{q}, \dot{\hat{q}} \rangle}^{=0} + r^2 \left| \dot{\hat{q}} \right|^2 \right) - \left( f(\hat{q}) + \frac{k}{2} r^2 \right) \right) + \lambda(t)(|\hat{q}|^2 - 1)$$

$$= e^{\frac{\mu}{m} t} \left( \frac{m}{2} \dot{r}^2 + \left( \frac{m}{2} \left| \dot{\hat{q}} \right|^2 - \frac{k}{2} \right) r^2 - f(\hat{q}) \right) + \lambda(t)(|\hat{q}|^2 - 1). \tag{26}$$

Now, we harness the covariant property of the Lagrangian formulation, i.e., it preserves the form of the Euler-Lagrange equations after change of coordinates, to obtain the radial and angular Euler-Lagrange equations,

**Radial Euler-Lagrange equation:** $\partial_r \mathcal{L} = \frac{d}{dt} \left( \frac{\partial \mathcal{L}}{\partial \dot{r}} \right)$,

$$m\ddot{r} + \mu \dot{r} = \left( m \left| \dot{\hat{q}} \right|^2 - k \right) r. \tag{27}$$

*Derivation.* We evaluate $\partial_r \mathcal{L} = \frac{d}{dt}\left(\frac{\partial \mathcal{L}}{\partial \dot{r}}\right)$ by substituting the Lagrangian (Eq. 26). By evaluating the left hand side, we get

$$\frac{\partial \mathcal{L}}{\partial r} = e^{\frac{\mu}{m}t}\left(m\left|\dot{\hat{q}}\right|^2 - k\right)r.$$

By evaluating the right hand side, we get

$$\frac{d}{dt}\left(\frac{\partial \mathcal{L}}{\partial \dot{r}}\right) = \frac{d}{dt}\left(me^{\frac{\mu}{m}t}\dot{r}\right) = \mu e^{\frac{\mu}{m}t}\dot{r} + me^{\frac{\mu}{m}t}\ddot{r}.$$

By equating the left and right hand sides and dividing both sides by $e^{\frac{\mu}{m}t}$, we obtain the radial Euler-Lagrange equation.

$\square$

**Angular Euler-Lagrange equation for $\hat{q}$:** $\partial_{\hat{q}}\mathcal{L} = \frac{d}{dt}\left(\frac{\partial \mathcal{L}}{\partial \dot{\hat{q}}}\right)$

$$m\ddot{\hat{q}} + \mu\dot{\hat{q}} = -\frac{1}{r^2}\nabla f(\hat{q}). \tag{28}$$

*Derivation.* We evaluate $\partial_{\hat{q}}\mathcal{L} = \frac{d}{dt}\left(\frac{\partial \mathcal{L}}{\partial \dot{\hat{q}}}\right)$ by substituting the Lagrangian (Eq. 26). By evaluating the left hand side, we get

$$\frac{\partial \mathcal{L}}{\partial \hat{q}} = -e^{\frac{\mu}{m}t}\nabla f(\hat{q}) + 2\lambda(t)\hat{q}.$$

By evaluating the right hand side, we get

$$\frac{d}{dt}\left(\frac{\partial \mathcal{L}}{\partial \dot{\hat{q}}}\right) = \frac{d}{dt}\left(e^{\frac{\mu}{m}t}m\dot{\hat{q}}r^2\right) = e^{\frac{\mu}{m}t}\left(\mu r^2\dot{\hat{q}} + mr^2\ddot{\hat{q}} + 2mr\dot{r}\dot{\hat{q}}\right).$$

By equating the both sides and rearranging terms, we get,

$$\dot{\hat{q}} = \left(1 + \frac{2m\dot{r}}{\mu r}\right)^{-1}\left(-\frac{1}{\mu r^2}\nabla f(\hat{q}) - \frac{m}{\mu}\ddot{\hat{q}} + \frac{2\lambda(t)}{\mu r^2}e^{-\frac{\mu}{m}t}\hat{q}\right).$$

Now, consider the limits of $m \to 0$ and $k \to 0$, which are justified in practice with the small learning rate $m \propto \eta$ and weight decay $k$. In this limit, the $\dot{r}$ becomes small

$$\dot{r} = \frac{1}{\mu}\left(m|\dot{\hat{q}}|^2 r - kr - m\ddot{r}\right) \sim O(m) + O(k),$$

and $m\dot{r} \sim O(m^2) + O(mk)$ will not be the leading order terms. Thus, by only keeping the first order terms in $m$ and $k$, we get

$$m\ddot{\hat{q}} + \mu\dot{\hat{q}} = -\frac{1}{r^2}\nabla f(\hat{q}).$$

$\square$

## C.2  Limiting dynamics

The derived Euler-Lagrange equations accurately describe deep learning dynamics in realistic settings with a finite learning rate $\eta$, momentum $\beta$, and weight decay $k$. Here, we demonstrate the power and accuracy of the Euler-Lagrange equations by deriving important results in the literature that have already been verified empirically. Unlike existing works using a discrete-time framework, our continuous-time framework eliminates all the tedious algebraic calculations and allows us to obtain the results immediately.

**Angular velocity.** First, by assuming a steady state of radial dynamics $\dot{r} = 0$ in equation Eq. 27, we immediately obtain a condition for angular velocity $m|\dot{\hat{q}}|^2 - k = 0$, which then yields a relationship

$$|\dot{\hat{q}}|^* = \sqrt{\frac{k}{m}} = \sqrt{\frac{2k}{\eta(1+\beta)}}. \tag{29}$$

By discretizing this continuous-time expression $|\hat{q}(t+\eta) - \hat{q}(t)| = \sqrt{\frac{2\eta k}{1+\beta}}$ we get a relationship that is consistent with the literature [77].

**Radius.** Similarly, by substituting a relationship from Eq. 28, $\dot{\hat{q}} = -\frac{1}{\mu r^2}\nabla f(\hat{q}) - \frac{m}{\mu}\ddot{\hat{q}}$, into Eq. 27 with the steady norm assumption, we obtain a condition for a steady radius $r^*$,

$$r^* = \sqrt[4]{\frac{m}{k\mu^2}}\sqrt{|\nabla f(\hat{q})|} = \sqrt[4]{\frac{\eta(1+\beta)}{2k(1-\beta)^2}}\sqrt{|\nabla f(\hat{q})|}, \tag{30}$$

where we kept only the leading order in $m$.

**Learning dynamics.** Plugging this expression obtained from the steady-state condition of Eq. 27 back into Eq. 28, we find those steady-state learning dynamics is driven by the normalized gradients scaled by intrinsic learning rate as,

$$m\ddot{\hat{q}} + \mu\dot{\hat{q}} = -\sqrt{\frac{k\mu^2}{m}}\frac{\nabla f(\hat{q})}{|\nabla f(\hat{q})|}, \tag{31}$$

consistent with results derived in [78] in the discrete-time setting.

## D   Implicit adaptive optimization by normalization layers

Here, we ignore the inertia term in Eq. 16, assuming that the mass (learning rate) is finite but small and the learning dynamics is in the overdamped regime. In this setting, the Eq. 16 reduces to,

$$\frac{d}{dt}r^2 = -\frac{2k}{\mu}r^2 + \frac{2m}{\mu^3 r^2}|\nabla f(\hat{q})|^2. \tag{32}$$

In the setting of gradient descent with no momentum ($\mu = 1 - \beta = 1$ and $m = \eta/2$), this expression simplifies to the one previously obtained with the *modified loss* approach in Eq. 21 of [1] and Eq. 35 of [46].

To solve this differential equation, first we multiply both sides by $2r^2$ to get,

$$2r^2\frac{d}{dt}r^2 = -\frac{4k}{\mu}(r^2)^2 + \frac{4m}{\mu^3}|\nabla f(\hat{q})|^2.$$

Rearranging terms and multiplying both sides by $e^{\frac{4k}{\mu}t}$ gives,

$$e^{\frac{4k}{\mu}t}\frac{d}{dt}(r^2)^2 + \frac{4k}{\mu}e^{\frac{4k}{\mu}t}(r^2)^2 = \frac{4m}{\mu^3}e^{\frac{4k}{\mu}t}|\nabla f(\hat{q})|^2.$$

By integrating both sides,

$$\left[(r^2)^2 e^{\frac{4k}{\mu}t}\right]_0^t = \frac{4m}{\mu^3}\int_0^t e^{\frac{4k}{\mu}\tau}|\nabla f(\hat{q}(\tau))|^2 d\tau,$$

and rearranging terms, we get

$$r^2(t) = \sqrt{\frac{4m}{\mu^3}\int_0^t e^{-\frac{4k}{\mu}(t-\tau)}|\nabla f(\hat{q}(\tau))|^2 d\tau + e^{-\frac{4k}{\mu}t}r^4(0)}. \tag{33}$$

# E Experimental details

All the experiments were run using the PyTorch code base.

**Dataset.** We used Tiny ImageNet dataset to generate all the empirical figures in this work. The Tiny ImageNet dataset consists of 100,000 training images at a resolution of $64 \times 64$ spanning 200 classes.

**Model.** We use standard VGG-11 models for all our experiments with a batch normalization layer between every convolutional layer and its activation function.

**Hyperparameters.** The key hyperparameters we used are listed with each figure. Unless specified otherwise, the standard hyperparameters we used are the momentum $0.9$, train batch size $128$, weight decay $10^{-4}$, where we drop the initial learning rate by a factor of ten after 30, 60, and 80 epochs.