# OpenReview forum: "Noether’s Learning Dynamics: Role of Symmetry Breaking in Neural Networks"
_NeurIPS.cc/2021/Conference — NeurIPS 2021 Poster_

### Official Review · Reviewer_6ME4 · 2021-07-11

**Rating:** 7
**Confidence:** 4

**Summary:**

The gradient descent with a momentum can be regarded as the Newtonian equation with damping. The output of the batch normalization is invariant to the scale of the input (i.e., a scale symmetry). When a BN is inserted after a convolution, the whole network is invariant to the weight scale. A typical gradient descent algorithm ignores (i.e., breaks) this symmetry. Thanks to this symmetry breaking, the learning rate is implicitly adapted, and the adaptation is equivalent to RMSProp's.


**Limitations And Societal Impact:**

The theoretical analysis is only applicable to the batch normalization, while the title seems to be more general. No negative societal impact.


**Main Review:**

Firstly, the main result is very interesting. The perspective from the symmetry breaking is unique and provides a new insight to the reason why BN works. The reviewer still has several questions about experiment and relationship with physics.

1) In the experiment, the VGG with BN is compared with a "constrained filter norm" variant, whose weight is normalized after each update. Even then, the gradient and the momentum term are calculated under the assumption of no constraints. This gap might affect the performance negatively. Hence, the performance degradation of the "constrained filter norm" variant might be insufficient to prove the effectiveness of the implicit adaptation by BN.

2) How necessary is the perspective of the Noether's theorem and symmetry breaking? While a physical analogy is well known to the GD with momentum, the equivalence between (17) and (18) might be derived without it.

3) Some prior works such as [34] are introduced in Section 5. In particular, [34] also provided the theoretical explanation of the reason why BN works. The relationship with and difference from prior works should be described in detail.

4) The title might be excessive advertising. The theoretical analysis does not describe the general role of the symmetry breaking in deep learning, but it is applicable only to BN

5) At line 62, the sentence of BN's symmetry might have a typo. Correctly, it might be BN((1+s)qx)=BN(qx).

**Time Spent Reviewing:**

6

---

> ### Author Response · Authors · 2021-08-10
> **Response to Reviewer 6ME4**
>
> Dear Reviewer 6ME4,
>
> Thank you for your thoughtful review, and we are delighted that you found "the main result is very interesting" and "The perspective from the symmetry breaking is unique and provides a new insight". In the following, we'd like to answer your thoughtful questions.
>
> ----
>
> **Q1.** Thank you for this question. We'd like to clarify that the gradient under BatchNorm layer does approximately respect the geometrical constraints by construction. When the parameter vector $q \in \mathbb{R}^N$ inputs to the BatchNorm layer, the loss function is invariant under scaling $a \in \mathbb{R}$ of the parameter vector $f(aq)=f(q)$. Differentiating both sides with respect to the scaling constant and evaluating it at identity $\partial_a f(aq)|_{a=1}=0$ shows that the gradient is always orthogonal to the parameter vector itself $\langle \nabla f(q), q \rangle = 0$. Importantly, this orthogonality condition is equivalent to computing the gradient assuming that the parameter vector is restricted to the hypersphere in the limit of small step sizes. We agree with you that it would be an interesting future direction to impose the geometrical constraint more strictly by developing an optimizer that explicitly handles the effects of finite learning rates.
>
> ----
>
> **Q2.** Thank you for this thought-provoking question. Indeed, the two theoretical formulations based on differential equations of motion and variational principles are theoretically equivalent, just like how Newtonian and Lagrangian mechanics are. Thus, it must be possible to derive the specific result of Eq. (17) without the variational formulation. However, the variational formulation and Noether's theorem practically facilitate calculations and, more importantly, provides a unifying view to the problem as proven to be fruitful in physics.
>
> **Technical value facilitating the change of coordinates:** The Lagrangian formulation has a unique useful property that the form of equations of motion, the Euler-Lagrange equations, does not change when we transform coordinates. We have used this fact to derive the radial and angular Euler-Lagrange equations (see L. 273~276) by moving to polar coordinates more suitable to the geometry of scale symmetry.
>
> **Conceptual value offering a unified perspective:** We'd like to highlight that we can apply Noether's learning dynamics developed in Section. 1~3 to *any differentiable symmetries in the architecture*. This means that we can immediately derive the dynamics of Noether charges that correspond to the respective symmetries beyond BatchNorm. Examples of differentiable symmetries in deep learning include (i) the rescale symmetry due to the ReLU function without BatchNorm, (ii) the translation symmetry due to the SoftMax function, which plays an important role in transformer architectures, and (iii) the rotational symmetry due to word embedding used in natural language processing. Among them, the re-scale symmetry in the ReLU function without BatchNorm $a^{-1} q_2 ReLU (a q_1 x) = q_2 ReLU (q_1 x)$ (see L.31-39 for details) introduces a distinct kinetic asymmetry from that of BatchNorm. Thus, this perspective opens a door for future work asking whether the broken-symmetry-induced dynamics of the corresponding Noether charge, $\frac{d}{dt} (|q_2|^2-|q_1|^2)$, produces analogous beneficial effects for learning as in the case of BatchNorm. We believe that this kind of analysis may reveal the reason why ReLU activation is effective compared to sigmoid. Finally, we highlight that the perspective offers a new unifying design principle of symmetry-breaking that may inspire us to develop new architectures.
>
> ----
>
> **Q3.** Thank you for suggesting the clarification. Yes, as we mentioned in Sec. 5, Ref. [34]  by Arora, Li, and Liu is indeed an important prior study for our analysis in Sec. 4. We agree with you on the merits of discussing the relevance of these studies more in detail, and we clarify our contributions in the following.
>
> First, as we discussed in Sec.5, our analysis in Sec. 4 and Ref. [34] both take theoretical approaches to reach a common conclusion that BatchNorm plays a beneficial role by controlling the effective learning rate. However, as we explain below, our analysis is complementary and brings new perspective to Ref. [34].
>
> In the previous study, Ref. [34] performed a thorough mathematical analysis on convergence properties in the presence of BatchNorm assuming a simplified setup with no momentum or weight decay. Under this assumption, the effective learning rate of the network with BatchNorm decreases monotonically, even when the true learning rate is kept fixed. (We can also see this by substituting $\beta=0$ and $k=0$ into our Eq. (17).)
> Grounding this intuition theoretically, Ref. [34] proved that in the presence of BatchNorm, even without adjusting the learning rate, the network converges to a stationary point in the rate of $T^{1/2}$ in $T$ iterations, asymptotically matching the best bound for gradient descent with well-tuned learning rates.
>
> In contrast, our analysis in Sec. 4 aimed to derive an exact solution to the time evolution of the effective learning rate in a practical setting with momentum and weight decay. To do so, we developed a continuous-time, Lagrangian formulation of the learning dynamics that allows us to use the basic tools of calculus, such as differentiation and integration. As a result, our theory has yielded a seemingly complex integral solution, Equation (17) via Noether's theorem. We then simultaneously developed a continuous-time model of the adaptive optimizer to find the equivalence between RMSProp and "BatchNorm+SGD+momentum+weight decay".  As a result, this non-trivial correspondence reveals the mechanisms of how learning rate, momentum, and weight decay together lead to successful learning. The novel predictions of our theory are summarized in "L.319-328", and these contributions remain novel in light of the previous works.
>
> To reflect your valuable feedback, we will (i) add the detailed discussion above to the related work section, and (ii) add citations in Sec. 4 to further clarify our technical contributions.
>
> ----
>
> **Q4.** Thank you for sharing a valuable perspective. We agree with you that it is important for the title to accurately represent the results. In the following, we'd like to explain our reasoning for choosing the title, and we are happy to modify the title to read "Role of Symmetry Breaking in Deep Learning", instead of "**The** Role of...", if you think this better reflects our results. First, we'd like to clarify how our theoretical analysis in Sec. 4. on BatchNorm generally applies to almost all modern deep learning systems. Successful training of modern deep networks requires stable forward/backward propagations of information throughout deep layers.  To solve this issue, normalization layers are ubiquitous building blocks of almost any deep learning system. While we have emphasized the concrete and practically important example of BatchNorm in our submitted draft, our theoretical analysis immediately applies to almost any normalization layers due to their scale symmetry. The examples include [InstanceNorm](https://arxiv.org/abs/1607.08022), [GroupNorm](https://arxiv.org/abs/1803.08494), [WeightNorm](https://arxiv.org/abs/1602.07868), and [scaled Weight Standarization](https://arxiv.org/abs/2101.08692) popular in vision models, as well as [LayerNorm](https://arxiv.org/abs/1607.06450) essential in natural language processing. In addition, we can generalize and apply our theory to layers beyond normalization layers as discussed in our reply to Q2, making our theory not limited to BatchNorm and generally relevant to deep networks.
>
> ----
>
> **Q5.** Yes, you are correct. Thank you for carefully spotting this typo.
>
> ----
>
> **Summary:** Thank you again for your thoughtful questions. As clarified above, your questions have helped us better contextualize our contributions. Although we did not emphasize it in the submitted draft, the application of our theory is not limited to BatchNorm. Rather, the very reason why we studied Noether's theorem in deep learning was to unify broad classes of symmetries ubiquitous in the deep learning architectures. Noether’s learning dynamics (results developed in Sec. 1,2,3) are applicable to any differentiable symmetries in the architecture, and the analysis in Sec. 4 can be immediately applicable to an array of normalization layers beyond BatchNorm. In future works, it would be fruitful to generalize our analysis on adaptive optimization (Sec. 4) to broader classes of differentiable symmetries such as the re-scale symmetry of the ReLU function. We are now ready to implement these clarifications to reflect your comments.

---

> > ### Comment · Reviewer_6ME4 · 2021-08-11
> > **Thank you for response.**
> >
> > Thank you very much for your thorough response. I raised my score. A removal of "The" from the title is fine to me.

---

### Official Review · Reviewer_L9t2 · 2021-07-16

**Rating:** 8
**Confidence:** 3

**Summary:**

The paper considers the role of symmetries in the learning dynamic by considering a continuous time formulation of the learning process in Lagrangian formalism. In such formulation there are "kinetic" and "potential" components to the dynamics, the former arising from the learning rules employed during optimization and the latter generated by the loss function landscape. Authors argue that for a large class of practically relevant network architectures there are symmetries that are present in the "potential" component, but not in the "kinetic" part.

Starting with Bergman Lagrangian formulation of the learning dynamics, authors derive the equations for time evolution of the Noether's charge associated with symmetry transformation (assumed to be present) in the network architecture.

For the case on scale symmetry introduces by BatchNormalization, authors observe that in case of training with gradient descent + momentum + weight decay; the training dynamics effectively reduces to a learning dynamics with an adaptive scale where the scale is related to the associated Noether's charge. This suggests that kinetic asymmetry induces an implicit adaptive property to the optimization symmetry. A connection between RMSProp and implicit adaptation is established based on continuous time learning model.

Lastly, authors test theoretical predictions by training a set of models verifying theoretical predictions on Tiny ImageNet dataset. For comparison, the adaptive rescaling effect is artfully removed by renormalizing the parameters throughout training.

**Limitations And Societal Impact:**

The work does not have negative societal impact.

**Main Review:**

The role of symmetries in deep learning is an active research area. The paper studies a relevant problem of investigating how symmetry properties of the model, combined with asymmetry of the learning rules, give rise to non-trivial adaptive learning dynamics. To the best of my knowledge the presented results are novel and interesting. Connections between architecture and model components are often highly important in practical settings for guiding the design decisions and understanding the interactions between architectures and optimization.

The paper is clearly written and cites relevant related works. One caveat related to the presentation of the related work is that it does not acknowledge the fact that Noether's theorem has been used in recent works (e.g. reference [14]). The hypotheses are adequately tested. The work reads complete and insightful.


**Time Spent Reviewing:**

6

---

> ### Author Response · Authors · 2021-08-10
> **Response to Reviewer L9t2**
>
> Dear Reviewer L9t2,
>
> Thank you for carefully understanding, thoroughly summarizing, and positively evaluating our work as: “results are novel and interesting”, “clearly written and cites relevant related works”, “The hypotheses are adequately tested.", and "The work reads complete and insightful.”
>
> We thank you also for pointing out the need to clarify the relationship with the reference. Indeed, Ref. [14] mentions that some of their results under gradient flow are "*akin/similar to Noether's theorem*". However, Ref. [14] do not use Noether's theorem, nor make formal connection. (We needed to reformulate the entire theory in the Lagrangian form to establish the connection!) Thus, our theory remains a novel application of Noether's theorem to the deep learning dynamics. We will clarify this point in the related work section to reflect your valuable feedback.

---

### Official Review · Reviewer_YvtG · 2021-07-16

**Rating:** 8
**Confidence:** 2

**Summary:**

The authors identify a "kinetic asymmetry" in the learning dynamics under gradient flow, where the kinetic energy is not invariant to the symmetries of the potential energy. An example is scaling: even when the network function is invariant to scaling or ReLUs or batch norm inputs, the learning dynamics are not. They derive a genealized version of Noether's theorem which takes this asymmetry into account and use it to study the implict adaptive optimization exhibited by batch normalization, showing that it is the same as that of RMSprop.


**Limitations And Societal Impact:**

Yes.

**Main Review:**

The topic and techniques of this paper are well outside of my comfort zone so I can only provide a high-level overview. From this high level, I find the premise of the paper exciting. The interaction between Lagrangian dynamics and optimization in machine learning is not new (e.g., the cited work of Wibisono, Wilson, and Jordan), but connections to  symmetry breaking and Noether's theorem, at least in the context of deep learning, seem to be original. The demonstrated consequence that the dynamics induced by batch normalization are the same as those of RMSprop is very cute. The paper is clearly written (although with quite a few misprints) and the provided examples are helpful.

My main objection is towards the generality of results and the corresponding claims. For example, the statement that "kinetic symmetry breaking is a key design principle to the success of a learning system" could be mollified. Symmetry breaking is certainly a beautiful perspective but it is one out of many.

While the theory and the symmetry-broken Noether's law hold generally, the only example in the paper is the scale symmetry exhibited by ReLU activations and batch norm (since scaling does not preserve kinetic energy). Could the authors suggest or speculate other interesting symmetries in deep learning that do not preserve kinetic energy? Another question is about using your insights opportunely---since learning dynamics is not invariant under symmetries, symmetries may be used to improve performance or accelerate training. On a related note: is there a connection to quiver representations of neural nets https://arxiv.org/abs/2007.12213, https://arxiv.org/abs/2107.02550?


**Time Spent Reviewing:**

3-4

---

> ### Author Response · Authors · 2021-08-10
> **Response to Reviewer YvtG**
>
> Dear Reviewer YvtG,
>
> We appreciate your thoughtful and positive feedback, summarized as *``I find the premise of the paper exciting.''* and *``The paper is clearly written.''* Thank you also for your question about the generality of our theory. In fact, our main result, Noether's learning dynamics [Eqs. (10), (11)] is immediately applicable to ***any differentiable symmetry***, without being limited to the scale symmetry of BatchNorm. However, as you pointed out, we focused on the analysis of BatchNorm in the submitted manuscript. This was to demonstrate how our theory yields practically important implications using a concrete and important example. In the following, we will resolve your concerns by showing that our theory applies to a wide range of deep learning architectures besides BatchNorm.
>
> ----
>
> > the statement that "kinetic symmetry breaking is a key design principle to the success of a learning system" could be mollified. Symmetry breaking is certainly a beautiful perspective but it is one out of many.
>
> We agree with you that "*symmetry breaking is certainly a beautiful perspective but it is one out of many*", and this message is what we intended to deliver throughout the manuscript. Although not in the abstract, we review complementary design principles in "Section 5. Related Works: Geometry of Optimization". In fact, the related works mostly demonstrate how *symmetry* helps optimization, and we think of our finding on the role of *symmetry breaking* as an interesting counter example. To reflect your feedback, we will revise the sentence in the abstract to read "*kinetic symmetry breaking is* **one of the** *key design principles*".
>
> ----
>
> **Our theory applies to any differentiable symmetry, including interesting ones:** In addition to the scale symmetry of BatchNorm, examples of differentiable symmetries in deep learning include (i) the rescale symmetry due to the ReLU function without BatchNorm, (ii) the translation symmetry due to the SoftMax function, which plays an important role in transformer architectures, and (iii) the rotational symmetry due to word embedding used in natural language processing. Among them, the re-scale symmetry in the ReLU function without BatchNorm $a^{-1} q_2 ReLU (a q_1 x) = q_2 ReLU (q_1 x)$ (see L.31-39 for details) introduces a distinct kinetic asymmetry from that of BatchNorm. Thus, it will be interesting to ask whether the broken-symmetry-induced dynamics of the corresponding Noether charge, $\frac{d}{dt} (|q_2|^2-|q_1|^2)$, produces analogous beneficial effects for learning as in the case of BatchNorm. In addition there are many normalization layers beyond BatchNorm with the scale symmetry as well, where the examples include [InstanceNorm](https://arxiv.org/abs/1607.08022), [GroupNorm](https://arxiv.org/abs/1803.08494), [WeightNorm](https://arxiv.org/abs/1602.07868), and [scaled Weight Standarization](https://arxiv.org/abs/2101.08692) popular in vision models, as well as [LayerNorm](https://arxiv.org/abs/1607.06450) essential in natural language processing. Finally, we highlight that *the ultimate goal of our work is to inspire new architectures building on the identified design principle of kinetic asymmetry*!
>
> ----
>
> **Can symmetries improve performance or accelerate learning?:** Yes. In fact, it has been much more common to study the beneficial role of symmetry rather than broken-symmetry, as we summarized in the related works, "Sec. 5. Geometry of optimization" (L352-360). Examples include second order optimization methods such as the natural gradient descent by S. Amari [10] and more recent efforts to enforce symmetries to deep learning dynamics such as the Path-SGD by B. Neyshabur et al. [2]. In this context, our work provides an interesting counter-example, where breaking the symmetry helps learning. We are also grateful for the pointers to quiver representations. It certainly is an interesting future direction to think about the connection!
>
> ----
>
> **Summary:** We thank you for your valuable feedback, and we hope that our replies above have clarified your concerns on the generality of our theory.

---

> > ### Comment · Reviewer_YvtG · 2021-09-02
> > **thank you for the clarifications**
> >
> > I'd like to thank the authors for a thorough and enlightening response. I will raise my score by one point.

---

### Official Review · Reviewer_ngQE · 2021-07-19

**Rating:** 7
**Confidence:** 3

**Summary:**

Based on the Euler-Lagrangian framework, this manuscript develops the theoretical method to model the learning dynamics of DNNs as symmetry breaking of the likelihood by the kinetic energy.
The model is expressed as the time evolution of Noether-current.
As a result, the author found that the learning dynamics of batch normalization are similar to RMSProp from the viewpoint of symmetry breaking.
Their findings deepen our understanding of batch normalization.
This study is unique not only because of the findings on batch normalization mechanics but also because it links the symmetry-breaking process of the likelihood function with the learning theory.

**Limitations And Societal Impact:**

I have described the concerns about limitation in Main Review.

**Main Review:**

The study about elucidating implicit adaptive optimization from the viewpoint of symmetry is excellent and unique, so this manuscript has originality and significance worthy of publication.
On the other hand, I can not understand some points related to the effectiveness of the theory as below.

The first concern is that there may be few cases in DNNs where the symmetry of the potential function is known.
For example, in their theory building on BatchNorm, the authors state:
"We investigate this physical system because the learning dynamics
264 of the most successful deep learning systems are exactly analogous to this system. For example, BatchNorm layers, ubiquitous in modern successful deep learning architectures, introduce scale symmetry in the parameter space q of the potential (loss) function."
However, I can not understand that BatchNormalization introduce scale invariance when it is subjected to an activation function such as sigmoid? (I can understand this for if the parameters are the regression coefficients of linear regression.)
Also, I understand that the space q is a parameter of the BatchNorm layer when considering the dynamics of BatchNorm.
And I understand that for the parameters of the other layers, the potential function is not symmetric to the scale transformation of these parameters.
Therefore, the symmetry of the parameters of the other layers is unknown.
From these understandings, I think it does not seem appropriate to compare the dynamics of RMSProp with those of DNN, since the space q in RMSProp is all the parameters of DNN.
By the way, isn't "f((1+s)q)=q" in L263 "f((1+s)q)=f(q)"?

The second is about the difficulty of the Lagrangian approach.
In general, it seems difficult to construct a Lagrangian that models the dynamics of a dissipative system.
I understood that the Bregman Lagrangian was constructed based on the Lagrangian of a known dissipative system called damped oscillation.
Therefore, when the time evolution of the error has a plateau, which is often the case in DNN training, the learning dynamics cannot be a damped oscillation, and that is why Bregman Lagrangian cannot be used.
Therefore, I could not understand the validity of theoretical analysis assuming damped oscillations.


There are some minor concern as below:


Around L168: Since q is supposed to be a generalized coordinate system, I think that authors should not make the assumption that $q = \dot{q}$ and $Q = \dot{Q}$.
In that case, I think that the premise of the discussion in this vicinity is no longer valid.

L119: There is no definition of bra-ket which represents the inner product.

L238: ",," might be typo

L275: "as$m" might be "as $m"

L295: "| as" might be "as"

L331, L332: There is no apparent definition of "unconstrained filter norm" and "constrained filter norm".
                    I think it is friendly for reader to describe these definition more clearly.

**Time Spent Reviewing:**

12hours

---

> ### Author Response · Authors · 2021-08-10
> **Response to Reviewer ngQE**
>
> Dear Reviewer ngQE,
>
> Thank you so much for taking the time to carefully understand our theory and write a thoughtful review that we find very helpful! We are pleased to hear that you found our results “excellent and unique” and you believe “this manuscript has originality and significance worthy of publication.” In the following, we'd like to resolve your concerns and explain how we plan to reflect your valuable feedback to make our theory more accessible.
>
> ---
>
> > *The first concern is that there may be few cases in DNNs where the symmetry of the potential function is known. ... I can not understand that BatchNormalization introduce scale invariance when it is subjected to an activation function such as sigmoid? ... I understand that for the parameters of the other layers, the potential function is not symmetric to the scale transformation of these parameters. Therefore, the symmetry of the parameters of the other layers is unknown.
>
>
> Thank you for your thoughtful questions to clarify the applicability of our theory. In the following, we will resolve your concerns by explaining how our theory applies broadly to almost any layer of modern deep neural networks, regardless of the non-linear activation function used with the normalization layers.
>
> **BatchNorm introduces scale invariance with any activation functions:** We usually apply BatchNorm right after the linear transformation of preactivation $x \in \mathbb{R}^{N^{[L-1]}}$ at layer $L-1$ by a weight matrix $W \in \mathbb{R}^{N^{[L]} \times N^{[L-1]}}$, *before* the non-linear activation function $g$ as described in [the original BatchNorm paper](https://arxiv.org/abs/1502.03167) by S. Ioffe and C. Szegedy (Ref. [4]). Thus, the activation vector of the following layer $z \in \mathbb{R}^{[L]}$ is calculated as $z = g (\text{BN}(Wx))$. Since the output of BatchNorm $\text{BN}(qx) = \frac{Wx - \text{E}[Wx]}{\sqrt{\text{Var}[Wx]}}$ is invariant under scaling of the weights $W\rightarrow aW$ by a constant $a \in \mathbb{R}$, the input to the activation function remains invariant under the scaling of the parameteres. Thus, the activation vector $z = g(\text{BN}(aWx))=g(\text{BN}(Wx))$ and ultimately the loss function $f(aW) = f(W)$ are scale invariant regardless of the activation function $g$. Therefore, the answer to your question, "*BatchNormalization introduces scale invariance when it is subjected to an activation function such as sigmoid?*", is yes.
>
> **Modern deep networks have normalization layers in almost every layer:** You are right that layers without BatchNorm are not necessarily scale symmetric. However, we'd like to highlight that in modern deep neural networks, almost all layers have a normalization layer to stabilize the forward/backward propagation across the deep layers. Concrete examples are available, for example, at standard implementations of vision models in PyToch ([Torch vision models](https://pytorch.org/vision/stable/models.html#torchvision.models.vgg16_bn)), where BatchNorm is applied wherever the convolutional layers are (see e.g., VGG or ResNet models). In this sense, BatchNorm layers are as ubiquitous as convolutional layers, which without a question are fundamental building blocks of modern deep networks. Practically, if we are using VGG-16, parameters in 15 layers out of the 16 layers, except for the last classification layer participate in scale symmetry. The same is true for ResNet-34, where parameters in 33 layers out of the 34 layers participate in Noether's learning dynamics studied in our analysis!
>
> **Our theory widely applies to many other normalization layers besides BatchNorm:** In Section 4. we focused on a specific example, BatchNorm. This is because we wanted to give a concrete example of how Noether's learning dynamics can be applied to derive insights into a practically important problem in modern deep learning. However, our analysis is not limited to BatchNorm, and can immediately be applied to other types of normalization layers widely used in modern deep learning. The examples include [InstanceNorm](https://arxiv.org/abs/1607.08022), [GroupNorm](https://arxiv.org/abs/1803.08494), [WeightNorm](https://arxiv.org/abs/1602.07868), and [scaled Weight Standarization](https://arxiv.org/abs/2101.08692) popular in vision models, as well as [LayerNorm](https://arxiv.org/abs/1607.06450) essential in natural language processing.
>
> **Even parameters without scale symmetry often participate in other types of differentiable symmetries:** Finally, we address the question of "What can Noether's learning dynamics tell us about layers without scale symmetry?" Here, it's important to note that Noether's learning dynamics developed in Sec. 1,2,3 are applicable to any differentiable symmetries that we can find in the architecture. For example, this practically implies that the last classification layer of the vision models, which does not respect scale symmetry, instead has translation symmetry due to SoftMax activation. Thus, we can still apply Noether's learning dynamics to conduct a similar analysis to the last layer as well. This makes Noether's learning dynamics applicable to all of the layers in modern deep networks!
>
> **Conclusion:** Thank you so much again for your thoughtful questions. Your questions motivated us to better clarify the applicability of our theory, and we will reflect your feedback in revision by adding the above discussion as a new paragraph.
>
> ---
>
> > *The second is about the difficulty of the Lagrangian approach. ... I understood that the Bregman Lagrangian was constructed based on the Lagrangian of a known dissipative system called damped oscillation...
>
> Thank you for this question. As we clarify in the following, we *do not* need to assume the damped oscillation dynamics to construct a Lagrangian, and thus this resolves your concern. We understand that this Lagrangian formulation is not yet a common tool in deep learning theory and will add more detailed background explanations and derivations.
>
> First, the Lagrangian formulation does not assume damped oscillator dynamics. In physics, damped oscillator dynamics refers to $m \ddot{q} + \mu\dot{q} + k q =0$, where $m$ is the mass of a particle, $\mu$ is the damping coefficient, and $k$ is the spring constant. As you implied, the crucial assumption of this damped oscillator model is that the potential function is quadratic $\propto k q^2$, which is incompatible with the transient learning dynamics (such as plateau) as you pointed out. While the dynamics resembles our models, the crucial difference is that the dynamics we study (e.g. Eq. (3) $\frac{\eta}{2}\ddot{q} + (1-\beta) \dot{q} + \nabla f(q) + kq =0$) incorporate *extra forcing term $\nabla f(q)$*. This forcing term originates from the full loss function $f(q)$ *without any simplifying assumption*, and thus our Lagrangian formulation does not make any major assumptions. In fact, we also highlight that **there is no additional assumption needed when moving to the Lagrangian formulation**. This is why our theory yielded practically important and accurate predictions as verified in Sec. 4.
>
> ---
>
> **Notations:** First, we’d like to make sure that we understand your concern accurately. When you wrote "*assumption that $q=\dot{q}$ and $Q = \dot{Q}$*", did you actually mean "$q = Q$ and $\dot{q} = \dot{Q}$"? We proceed assuming that this is the case. If so, in L165-166, we declared that “For the remainder, we always assume that any derivative of Q is evaluated by the identity s = 0” to keep the notations simple. Based on this notation, we can rewrite the above relationship explicitly as $q = Q_{s=0}$ and $\dot{q} = \dot{Q}_{s=0}$, which is true by definition. In revision, we will make notations more explicit to avoid such confusion.
>
> Finally, thank you so much for carefully spotting multiple typos and ambiguity of definitions! We have now fixed all of them and the changes will be in the revised manuscript.
>
> ---
>
> **Summary:** Thank you again for carefully going through our theory and pointing out important clarifications that were helpful! As we have shown, the results of our analysis apply broadly to almost all the layers in modern deep learning systems, and we don't need to make any additional assumptions in moving to the Lagrangian formulation. In revision, we will reflect your valuable feedback to communicate these intricate, but important theoretical backgrounds to make the manuscript more accessible.

---

### Author Response · Authors · 2021-08-10
**Response to all Reviewers**

Dear Reviewers,

We would like to thank the reviewers for the insightful feedback and for unanimously supporting the acceptance of our work as “The work reads complete and insightful.”[R L9t2], "very interesting"[R 6ME4], “I find the premise of the paper exciting."[R YvtG], and "this manuscript has originality and significance worthy of publication”[R ngQE]. We are pleased that all the reviewers found our main result, which intertwines broken symmetry and Noether's theorem, fundamental concepts in modern physics, with the practice of deep learning, to be "unique and excellent"[R ngQE], "very cute"[R YvtG], "novel and interesting"[R L9t2], and "unique and provides a new insight"[R 6ME4].

To reflect valuable feedback from the reviewers, we plan to revise our manuscript to include:
*  demonstrations of how our theory generally applies to any differentiable symmetry in deep learning architectures, beyond the example of BatchNorm.
*  detailed introductions to relatively fresh tools in deep learning theory, such as the Lagrangian formulation and differentiable symmetry.
* discussions of the broader implications of Noether's theorem for deep learning in light of existing works.

We reply to comments and questions from the reviewers individually in the following.

Sincerely,

---

### Decision · Program_Chairs · 2021-09-27

**Decision:**

Accept (Poster)

**Comment:**

Following in the tradition of physics, the paper considers the implications of differentiable symmetries exhibited by a given learning task for the dynamics of the associated learning process. Although similar ideas have previously appeared in the literature ([14], in particular), an explicit study of the suggested Lagrangian framework is conceptually new. The reviewers found this framework interesting and insightful, and appreciated the unique perspective it provides on the connection between batch normalization and the popular optimization method RMSProp.